# Proprotein Convertase Subtilisin/Kexin Type 9, Angiopoietin-Like Protein 8, Sortilin, and Cholesteryl Ester Transfer Protein—Friends of Foes for Psoriatic Patients at the Risk of Developing Cardiometabolic Syndrome?

**DOI:** 10.3390/ijms21103682

**Published:** 2020-05-23

**Authors:** Julita Anna Krahel, Anna Baran, Tomasz W. Kamiński, Iwona Flisiak

**Affiliations:** 1Department of Dermatology and Venereology, Medical University of Bialystok, Zurawia 14 St., 15–540 Bialystok, Poland; aannabaran@wp.pl (A.B.); iflisiak@umb.edu.pl (I.F.); 2Department of Pharmacodynamics, Medical University of Bialystok, Mickiewicza 2c St., 15–222 Bialystok, Poland; tomasz.kaminski@umb.edu.pl; 3Pittsburgh Heart, Lung and Blood Vascular Medicine Institute, University of Pittsburgh, Pittsburgh, PA 15260, USA

**Keywords:** psoriasis, metabolic syndrome, atherosclerosis, PCSK9, ANGPTL8, CEPT, SORT1

## Abstract

Psoriasis is a systemic, immune-metabolic disease with strong genetic predispositions and autoimmune pathogenic traits. During psoriasis progression, a wide spectrum of comorbidities comes into play with the leading role of the cardio-metabolic syndrome (CMS) that occurs with the frequency of 30–50% amongst the psoriatic patients. Both conditions—psoriasis and CMS—have numerous common pathways, mainly related to proinflammatory pathways and cytokine profiles. Surprisingly, despite the years of research, the exact pathways linking the occurrence of CMS in the psoriasis population are still not fully understood. Recently published papers, both clinical and based on the basic science, shed new light into this relationship providing an insight into novel key-players proteins with plausible effects on above-mentioned interplay. Taking into account recent advances in this important medical matter, this review aims to discuss comprehensively the role of four proteins: proprotein convertase subtilisin/kexin type-9 (PSCK9), angiopoietin-like protein 8 (ANGPLT8), sortilin (SORT1), and cholesteryl ester transfer proteins (CEPT) as plausible links between psoriasis and CMS.

## 1. Introduction

Psoriasis is a common, disfiguring, and stigmatizing immune-metabolic skin disease affecting approximately 2–4% of the world population [1,2]. In history, psoriasis was considered as a solely dermatological condition altering the skin, nails, and joints with unexplained pathophysiology. Since 2000, there has been a rapid rise in the pairing of psoriasis with the immune system and metabolic syndrome, which has led scientists to identify psoriasis as an immune-metabolic disease. Psoriatic patients tend to develop metabolic syndrome (MetS), including abdominal obesity, cardiometabolic diseases (CMDs), diabetes mellitus (DM), dyslipidemia, and non-alcoholic fatty liver disease (NALFD) [3]. Today, many factors lead to the occurrence and progression of the disease, namely, genetic predisposition, lifestyle, viral and bacterial infections, and numerous medications used in cardiology and immunology [1,4]. The exact etiology and molecular background of psoriasis have not been dealt with in-depth, but recent years have produced abundant new clinical findings that clarified part of psoriasis pathophysiology.

First, the innate and adaptive immune responses and cytokines-dependent mechanisms are considered fundamental pathological processes priming the occurrence and severity of the disease. Inflammation is the immune system’s response to harmful stimuli, such as pathogens, damaged cells, toxic compounds, or irradiation. In general, a lasting, pro-inflammatory state is found in various conditions, including atherosclerosis, obesity, and psoriasis [1]. Acute and chronic phases of inflammatory process have been linked to increased morbidity of cardiovascular disease, neurological disorders, different types of cancer, and higher risk of deaths from these conditions. Interestingly, studying the plethora of different molecular and genomic pathways related to inflammatory processes resulted in the identification of pathways that are common for both, psoriasis and CMS. Considering genetic approach, alterations at the transcription levels of numerous genes, namely, renin, cytotoxic T-lymphocyte antigen 4 (CTLA4), and Toll-like receptor 3 (TLR3), which play a major role in the progression of both diseases have been identified [5]. Moreover, ongoing research investigates the roles of interleukins IL-12β and IL-23 as highly suspected players in psoriasis orchestration. Development of psoriatic symptoms has also been tied to lipid metabolism, which includes insulin resistance (IR), atherosclerosis, angiogenesis, oxidative stress, proatherogenic lipid and lipoprotein profile, and abdominal adipose tissue accumulation [4,6]. Next to lipid metabolism abnormalities, adipose tissue has been found to play a major role in psoriasis and CMS by serving as a critical source of diverse proinflammatory cytokines and adipokines. Recent studies published by Wolk and Kiluk and colleagues point out that this type of tissue releases molecules directly associated with interplay between CMS and psoriasis: TNF-α (Tumor necrosis factor α), IL-6 (interleukin 6), leptin, resistin, vaspin, and omentin [7,8]. The augmentation of the inflammatory response leads to the development of IR, lipid metabolism disturbances, vascular dysfunctions, and finally atherosclerosis [9]. At the same time, those disturbances lead to enhancement of adipose tissue metabolism, rebounded inflammatory processes, and acceleration in psoriatic and CMS forming and progression. Concurrently, disturbing the lipid balance and augmented inflammatory response lead to NAFLD that is present in 50% of psoriatic patients and is closely related to CMS [10]. Due to its perpetual and inevitable character, this process is called “psoriatic march” and is shown in Figure 1.

Psoriasis and CMS interpenetrate each other mainly in a dyslipidemia-driven manner [11]. Several recent reports have pointed out that patients with psoriasis have been more frequently diagnosed with proatherogenic lipoprotein profile, characterized by hyperglyceridemia, elevated plasma concentrations of low-density lipoprotein (LDL), and lowered high-density lipoprotein (HDL) concentrations [11]. Although it is still debatable whether lipid abnormalities are primordial or psoriasis-derivative, recent studies provide us with the evidence that both conditions drive each other to higher morbidity [12]. The well-established recent theory implies that proatherogenic lipid profile and psoriasis pathomechanisms are directly combined. It has been proven that psoriasis triggers increased mortality phenotype from cardiovascular events—mainly, myocardial infarction (MI) and prothrombogenic environment [13,14]. Another question arises as to what are the key players that make this connection so tight. To answer this question, researchers started to investigate for biomarkers of common pathogenetic pathways leading to the development of psoriasis and its cardiometabolic comorbidities.

This paper is an overview of recent advances in the biology of proteins, which seem to be missing links between psoriasis and CMS. Based on an extensive literature review and our experience, we chose four proteins with high capability to play a crucial role in the observed phenomena, namely proprotein convertase subtilisin/kexin type-9 (PSCK9), angiopoietin-like protein 8 (ANGPLT8), sortilin (SIRT1), and cholesteryl ester transfer proteins (CEPT). The study highlights their role in metabolic syndrome, cardiovascular comorbidities, immunological diseases, and potential cross-talking in psoriasis.

## 2. Proprotein Convertase Subtilisin/Kexin Type-9 (PCSK9)

Proprotein convertase subtilisin/kexin type-9 belongs to the group of serine proteinases entitled proprotein convertases (PCs). PCSK9 was originally discovered in 2003 by Seidah et al. [15] as a molecule that affects significantly lipid homeostasis, mainly LDL level in the blood and as a target for treating hypercholesterolemia. Since 2003, PCSK9 has been rigorously investigated in numerous studies in divergent fields of biology and medicine, which is now perceived as a pleiotropic protein with multiple properties toward the human body. The proprotein convertase superfamily spans a wide range of proteins—mainly, protein convertases 1, 2, 3, 5, 7, furin, paired basic amino-acid cleaving enzyme 4, and subtilisin kexin isozyme-1, which exert various biologically significant functions [16]. Members of the subtilisin/kexin like proprotein convertase (PCSK) family modify dormant prohormones into their terminal bioactive equivalents and stimulate assembly and liberation of cytokines, thereby maintaining natural homeostasis and playing an important role in inflammatory processes [17]. The acronym of the family—*PCSK*—was established as a result of their similar structure compared to the bacterial protease subtilisin and kexin in yeast [18]. PCSK9 gene is located on the short arm of chromosome 1 at position 32.3 and is formed of 12 exons and 11 introns. PCSK9 consists of three domains: the N-terminal prodomain, the catalytic domain, and the C-terminal domain [19]. Interestingly, PCSK9 was previously known as neural apoptosis-regulated convertase-1 (NARC1), since it was mistakenly thought that this protein exerts the same spectrum of biological interplay like other members of the PC family [20]. Years later, it was discovered that PCSK9 possesses a unique function and acts as a chaperone protein to LDL receptor (LDLR) leading to its degradation and thus increases plasma LDL levels [20].

PCSK9 is expressed predominantly in the liver; however, it has also been found to be expressed at different levels in the lungs, intestines, kidneys, brain, endothelial cells, and macrophages. This discovery laid solid grounds for inclusion this protein to the factors affecting cardiovascular homeostasis and hemostasis, inflammatory processes, and atherosclerosis [21]. There are two main regulators of PCSK9 gene expression: sterol-regulatory element-binding protein-2 (SREBP-2) and hepatocyte nuclear factor 1 (HNF1) [18]. Thus, circulating PCSK9 is synthesized as a soluble zymogen mainly in the liver, and its level is regulated by multiple factors: endogenous factors like hormones (including estrogen, insulin) and exogenous factors like diet, physical activity, and cardiological drugs [19]. Interestingly, its levels are higher in women than in men, and its concentrations decrease with age in men and increase along with maturing in women. The one and only PCSK9 substrate discovered so far is its prodomain: ProPCSK9. PCSK9 undergoes auto cleavage, which is an essential process for assembling the well-structured protein excretion, and then it is released into the blood through secretory pathways [21]. Circulating in the bloodstream, PCSK9 binds to the extracellular domains of transmembrane receptors like LDLR. The entire process leads to internalizations of the complex PCSK9-LDLR to the hepatocyte, and the further complex is subjected to lysosomal degradation [22]. Finally, a down-regulation of LDLR at the hepatocyte surface occurs, and increased circulating levels of LDL are observed [21]. The scheme of this process is shown in Figure 2.

Shortly after the principal paper describing PCSK9 by Seidah et al. [15] in 2003 was published, Abifadel et al. [23] discovered that gain of function (GOF) mutations within PCSK9 leads to the development of autosomal dominant hypercholesterolemia. In animal models, it has been demonstrated that direct inhibition of PCSK9 gene expression resulted in a decrease of LDL plasma concentrations in mice. In contrast, transgenic mice overexpressing PCSK9 showed a reduced number of LDLRs on the surface of the hepatocyte and significantly increased LDL levels [21]. Over the recent decade, PCSK9 became the subject of numerous studies focused on understanding the role of PCSK9 in diseases other than dyslipidemia.

Recently, many interesting papers have been published in this area. In 2016, Sucajtys-Szulc et al. observed that up-regulation of liver PCSK9 gene expression seems to be a possible cause of hypercholesterolemia in experimental chronic renal failure [24]. In the same year, Haas et al. [25] showed that nephrotic syndrome increases liver PCSK9 expression and its release in a murine model. The research conducted by Ozkan in 2015 proved that hypothyroidism increases PCSK9 expression, activity, and secretion in patients with thyroid cancer [26]. Interesting observations have been made by Costet et al. [27] They proved that hyperinsulinemia augments PCSK9 expression in mice, but PCSK9 level is simultaneously decreased in patients with type 1 diabetes compared to type 2 diabetes, suggesting a divergent role of the protein in insulin metabolism [27]. Moreover, NAFLD, which is closely linked to psoriasis morbidity, increases circulating PCSK9 concentrations, which in turn correlate positively with hepatic fat accumulation, irrespectively of other metabolic factors [28].

In 2019, Jeedundang investigated in a clinical manner PCSK9 levels and their association with metabolic parameters in 436 subjects in terms of the view of metabolic syndrome and menopausal processes. The study has shown that circulating PCSK9 level is increased in postmenopausal women with metabolic syndrome and suggested that this elevation might exacerbate the risk of MetS amongst those women [29]. Further, Fang et al. [30] investigated the level of PCSK9 in systemic lupus erythematosus (SLE) and found exaggeration in PCSK9 concentrations in patients with coexisting SLE. Interestingly, the increase was the highest in those with co-occurring lupus nephritis (LN) [30]. In the contrast to the above-mentioned data, in 2016, Ferraz-Amaro et al. [31] observed that patients with rheumatoid arthritis (RA) were shown with decreased PCSK9 concentrations than controls; however, this outcome might be influenced by simultaneously administrated tocilizumab IL-6 receptor blocker. Further studies provided that PCSK9 is also expressed in arterial wall cells, including endothelial cells, smooth muscle cells, and macrophages. Increased expression and activity of PCSK9 in those cells led to inflammation and further to atherosclerosis [19].

The putative involvement of PCSK9 in provoking and predicting cardiovascular events has been rigorously examined in both, clinical and basic science settings. A comprehensive meta-analysis and in-depth review performed by Vlachopoulos et al. [32] shows that PCSK9 concentrations are modestly but significantly associated with increased risk of total cardiovascular events. Obtained results suggest a predictive role of PCSK9 levels for the occurrence of cardiovascular diseases and support the possible clinical role of PCSK9 inhibitors. Furthermore, Zhi-Han et al. [33] found that PCSK9 seems to be a mediator of the inflammatory responses in the process of atherosclerosis, showing that the inactivation of the PCSK9 gene directly inhibits the formation of atherosclerotic plaques by reducing the severity of inflammation in blood vessels and inhibiting the toll-like receptor and NF-κB (nuclear factor kappa) light-chain-enhancer of activated B cells. Moreover, two independent studies proved that patients with acute coronary syndrome have been diagnosed with higher PCSK9 levels that were also associated with the severity of inflammation and overall prognosis [34,35].

Chronic inflammation is a key factor in the progression of atherosclerosis but is also commonly observed in psoriasis, where it is strongly associated with disease progression and morbidity. The close relationship between PCSK9 levels and inflammation was demonstrated in several studies. Work by Dwivedi et al. [36] demonstrates that PCSK9 deficiency extends protection against systemic bacterial dissemination, organ pathology, and tissue inflammation, particularly in the lungs and in the liver, while PCSK9 overexpression exacerbates multi-organ pathology as well as the hypercoagulable and pro-inflammatory states in early sepsis. Robust in vivo investigation reported amelioration of lipopolysaccharide (LPS)-induced inflammation and decreased serum levels of TNF-alfa, IL-6, and macrophages in PCSK9 knock-out mice [37]. Finally, the in vivo results were supported by the results of the clinical studies, in which patients with sepsis carrying a PCSK9 LOF (loss of function) allele had decreased plasma levels of proinflammatory cytokines such as TNF-α, IL-6, and IL-8 [37]. Moreover, Ranniko et al. [38] observed that PCSK9 is upregulated in the blood culture-positive infections in humans and presumed that plasma PCSK9 resembles acute-phase proteins; thus, its expression is induced during infection and correlates positively with CRP level. Additionally, PCSK9 concentrations were also observed to be significantly correlated with hs-CRP (high sensitivity C-reactive protein) in patients with the existence of cardiovascular disease [39]. This data highlights that PCSK9 should be considered as a pro-inflammatory factor interpenetrating different diseases that are characterized with the inflammatory background.

The straight consequence of revealing the impact of PCSK9 on lipid metabolism led to asking questions about its connection with CMS. In 2019, research showed that patients with stable coronary artery disease low PCSK9 plasma levels were associated with a particular metabolic phenotype (low HDL cholesterol, the metabolic syndrome, obesity, insulin resistance, and diabetes) [32].

This area of research represents an important starting point for the better understanding of the physiological role of PCSK9, also considering the recent approval of new therapies involving anti-PCSK9. Representatives of this new group of drugs are evolocumab and alirocumab, monoclonal antibodies against PCSK9. In the randomized clinical trial MENDEL-2, evolocumab (used in the dose of 140 mg 1–2 times a month) reduced LDL-C by 55–57% on average compared to placebo [40]. Studies with alirocumab also provide promising results: after 24 weeks of the ODDYSEY COMBO study, the PCSK9 inhibitor significantly reduced LDL-C (46%) compared to placebo group [41].

### PCSK9 and Psoriasis

PCSK9 has been recognized in the last years as the molecule that may be directly involved in the occurrence and progression of psoriasis. In 2011, Cao with colleagues [42] proved the existence of a cytokine-triggered regulatory network for PCSK9 expression that is linked to Janus Kinase (JAK) and the extracellular signal-regulated kinase (ERK) signaling pathway activation. ERK activity was observed during the increase in lesional psoriatic skin compared with nonlesional psoriatic skin, and clearance of psoriasis normalizes its activity [30]. Moreover, JAK inhibitors have been shown as high-potent anti-inflammatory agents that alleviate the course of psoriasis [42]. In turn, Zhi-Han Tang et al. [33] found that PCSK9 plays the role of a mediator of the inflammatory response in atherosclerosis. They showed that the suppression of the PCSK9 gene independently inhibits the formation of atherosclerotic plaque by reducing the concentrations of the circulating cytokines and inhibiting the TLR4 receptor (toll-like receptor 4) as well as nuclear factor kappa-light-chain-enhancer of activated B cells pathway activation (NF-κB). Both pathways—TLR4-dependent and NF-κB pathway—are essential in the pathogenesis of numerous dermatoses, including psoriasis [33]. Another link between PCSK9 and psoriasis may be mediated by adiponectin receptor agonists (AdipoR agonists) activation. Sun et al. [43] noted that AdipoR agonists induced peroxisome proliferator-activated receptor gamma (PPARs) expression, which plays a role as a PCSK9 expression regulator. It is well-established that PPARs-gamma plays an important role in psoriasis as anti-inflammatory factors that augment keratinocyte maturation, decrease proliferation, and induce apoptosis [44]. Simultaneously, oxidative stress that acts as pro-psoriatic factor comes into play, exacerbating the deteriorating effects of inflammation on endothelial cells, what in turn leads to the higher risk of atherothrombotic complications in psoriasis patients. Schluter et al. proved that the high concentration of oxidized LDL induces the expression of PCSK9 in endothelial cells and cardiomyocytes [45]. Moving further, the crosstalk between adiponectin, metabolic, inflammatory diseases, and psoriasis has been proven in many studies [46]. Interestingly, in 2019, the direct, unquestionable link between PCSK9 and psoriasis was proven by Luan et al. [47]. In that paper, the authors examined the potential of topically applying small interfering RNA targeting PCSK9 as the psoriasis treatment using murine and cell culture models. They proved that PCSK9 is overexpressed in psoriatic-like lesions and found that suppressing PCSK9 can reduce the inflammatory reaction induced by imiquimod through inhibition of hyperproliferation and apoptosis of keratinocytes. Additionally, the authors also investigated PCSK9 levels in lesions of psoriasis patients using quantitative RT-PCR assays, which have shown that PCSK9 expression was five times higher in psoriatic plaques compared to unaffected skin. Also, immunohistochemistry showed that psoriatic lesions had increased expression of PCSK9 in keratinocytes and epithelial cells of the vessels in the dermis. Additionally, they found that the combination of narrow-band-UVB and suppressing of PCSK9 might augment apoptosis of keratinocytes [47]. Moreover, in a recently published paper regarding our investigation, we confirmed that PCSK9 seems to be a novel marker of psoriasis and a putative explanation of lipid disturbances, which are common in patients with psoriasis and are vital for the further developing of metabolic syndrome [48]. To conclude, accumulated data suggest that PCSK9 plays a crucial role in psoriasis, promoting the occurrence of its comorbidities and it should be considered as an important and effective therapeutic target. Therefore, further investigations on the exact role of PCSK9 in psoriasis and its comorbidities are needed. Processes mentioned above have been shown in Figure 3.

## 3. Angiopoietin-Like Protein 8 (ANGPL8)

Angiopoietin-like protein 8, also known as betatrophin, lipasin, re-feeding induced fat and liver (RIFL), or chromosome 19 open reading frame 80 (C19orf80) belongs to the angiopoietin family-like proteins (ANGPTLs). Most of the family members show structural similarity to angiopoietins but do not have binding capabilities to their receptors, namely, EGF-like domain 1 (Tie1), TEK, and Tie2 (endothelial-specific receptor tyrosine kinase) [49]. It is well known that proteins belonging to the ANGPTLs family are characterized by pleiotropic effects due to involvement in the lipid and glucose metabolism. Recent papers highlighted its role in inflammation, however, the exact molecular targets of ANGPLs remain largely unrecognized and poor defined. Since the family’s discovery, eight proteins associated with the ANGPTLs family have been described and designated with the numbers 1–8 (ANGPTL1-8) [49]. ANGPTL8 was originally discovered in 2004 by Dong et al. as a hepatocellular carcinoma-associated antigen [50]. For several years, there has been a gap in investigations aimed to classify this molecule. Finally, in 2012, Ren et al. [51] demonstrated the existence of the connection between lipid metabolism and the presence of ANGPTL8 assembly in a rodent model. He found out that that triglyceride (TG) levels were reduced three-fold in mice lacking the ANGPTL8 gene. It shed new light into the biology of this protein and pushed researchers to investigate its molecular patterns deeply.

The gene encoding ANGPTL8 is located on chromosome 19p13.2 and in humans ANGPTL8 is expressed mainly in the liver, with a significantly lower percentage of the expression in fat tissue, brain, rectum, and heart [52]. In 2012, Zhang et al. [53] proved that ANGPTL3 and ANGPTL4 are critical regulators of blood lipids due to lipoprotein lipase (LPL) inhibition, suggesting that elevated levels of proteins from this family are predictors for developing atherosclerotic cardiovascular diseases. Another important research was published by Quagliarini et al., [54] where authors revealed that the overexpression of ANGPTL8 is independently correlated with increased TG level and simultaneously ANGPL3, suggesting that these two proteins may be coherent. Furthermore, the biology of ANGPTL8 is not only narrowed to lipid metabolism. Yi et al., [55] showed that a 17-fold increase in β-cell proliferation was found due to overexpression of ANGPL8 in mouse liver, through an undefined receptor. Moreover, they reported that mice with an increased level of ANGPTL8 revised glucose tolerance. It clearly shows the possible implication of ANGPTL8 to the formation of a proatherogenic phenotype that could easily promote the occurrence of cardiometabolic syndrome. After years, the results of several further reports were inconsistent and controversial. Thus, the ability of ANGPTL8 to stimuli beta-cell proliferation has been tentative for quite some time [56]. Finally, the scientific hesitancy has recently been established. The conclusion is that ANGPTL8 does not affect beta-cell expansion in humans [57].

Accumulated data have highlighted that increased levels of ANGPTL8 were observed in obesity, which is closely linked with CMS and also psoriasis. In 2019, Ye et al. [58] provided the evidence that increased ANPTL8 level may directly augment the risk of obesity in adults. Abu Fahra et al. [59] evaluated the levels of ANGPTL8 in obese and non-obese subjects before and after exercise training. They reported a positive correlation between levels of ANGPTL8 and BMI values as well as waist/hip ratio in non-diabetic subjects. Interestingly, the augmented levels were restored to physiological conditions after regular physical activities [59]. The same group showed that ANGPTL8 levels are elevated in patients with diagnosed CMS and this dependency was significantly associated with hs-CRP levels, highlighting its potential role in metabolic and inflammatory pathways that are crucial in the course of psoriasis. Another evidence linking ANGPTL8 with CMS is a paper investigating the ANGPTL8 in 556 diabetes with T2D and showing that ANGPTL8 level was three times higher in diabetic patients with T2D compared to controls [60]. The scientists also reported that ANGTPL8 was an independent predictor of T2D occurrence and was further correlated positively with factors promoting CMS including age, BMI, or waist/hip ratio [60]. Except for obesity, CMS, and T2D, it has been proven that ANGPL8 expression and activity might be influenced by thyroid function and polycystic ovary syndrome [60].

### ANGPL8 and Psoriasis

A presumed common link between ANGPTL8 and psoriasis seemed to be mediated by myokine called irisin. Irisin, a recently discovered myokine was initially isolated from muscle tissue by Bostrom et al. [61]. Irisin, a thermogenic adipomyokine produced by FNDC5 cleavage is involved in the browning of adipose tissue and is believed to play a role in insulin resistance mechanisms [62]. It has been found that that irisin augments the expression assembly of ANGPTL8 protein during adipocyte differentiation. Moreover, the positive correlation between the protein and irisin in persons with type 1 diabetes (T1D) has been found [62]. In 2017, Baran et al. [63] investigated the connection between irisin and psoriasis and reported that irisin serum levels were increased in psoriatic patients compared to healthy controls. The group did not find significant correlations between investigated myokine and indicators of metabolic disorders including BMI but in contrary significant positive correlations with high-sensitive C-reactive protein (hs-CRP) were noted. Presumably, it indicates that irisin may serve as a marker of inflammation state leading to CMS in psoriatic patients.

Data linking ANGPTL8 with psoriasis is based on NF-κB and ANGPTL8 interplay, first described by Zhang et al. [64]. The activation of NF-κB is mediated by TNF-α, which plays an unquestionable role in the pathogenesis of psoriasis. Zhang et al. reported that ANGPTL8 acts as a negative regulator in TNFα-triggered NF-κB activation, suggesting that the protein is a critical step to downregulate inflammatory responses [64]. The same group also assayed the circulating ANGPTL8 levels in the serum of patients with systemic inflammatory response syndrome (SIRS) and found that the levels were significantly increased, suggesting that ANGPTL8 diminishes acute phase of the inflammatory response [64]. Today, the exact role of ANGPTL8 in inflammatory processes and subsequently in the course of psoriasis is still poorly understood. Nevertheless, there is a strong background for further studies that may increase our knowledge of the interplay of ANGPTL8 and CMS inflammation in patients diagnosed with psoriasis.

## 4. Sortilin

Sortilin, comprising of 833 amino acids, is a high affinity, transmembrane, multi-ligand receptor, which belongs to the Vacuolar protein sorting 10 protein (Vps10p) domain receptor family [65]. It was described for the first time in 1997 by Petersen et al. as a receptor-associated protein (RAP) [66]. The Vps10p domain receptors family includes five members: sortilin, SorLA, sorCS1, sorCS2, and sorCS3. The gene of sortilin (*SORT1*) is located on chromosome 1p13.3 and is expressed predominantly in the central nervous system, hepatocytes, adipocytes, and macrophages. Interestingly, the 1p13 locus individually substantially affects the risk of development of coronary artery disease, thus promoting the occurrence of further CVD-related complications [67]. Furthermore, this locus is also associated with upregulation of serum LDL-C levels and seems to be a factor promoting atherosclerosis [68]. Musunuru et al. [69] proved that a common polymorphism at the 1p13 locus creates a transcription pattern that alters the hepatic expression of SORT1. This change makes its carriers more susceptible to LDL-C and VLDL metabolism disruptions and increases the risk of myocardial infarction. The SORT1 protein contains a Vps10p domain, which is a binding site for numerous ligands and takes a part in intracellular trafficking of protein between Golgi apparatus and the endosomes, transmembrane helix, and cytoplasmic tail [68]. An inactive precursor of SORT1 is synthesized in the endoplasmic reticulum, and further, it undergoes furin-mediated cleavage in the trans-Golgi network [70]. The highest concentration of SORT1 was found in Golgi structures, where it acts as a lysosomal sorting receptor. Interestingly, lower concentrations of this molecule were found on the plasma membrane, where the function of controlling the transmembrane transport via endocytosis has been proposed for SORT1 most often [70]. Furthermore, SORT1, based on the cell surface, may undergo transformation and, consequently, be released as a soluble form into the extracellular matrix [65].

The main function of sortlin is engaging in the lipid metabolism, mostly impacting lipid levels in the bloodstream and hepatic apoB lipoprotein metabolism. Sparks et al. [71] demonstrated that the reduction in SORT1 levels would be anticipated to increase circulating LDL, and thereby promote atherogenesis. As stated, the provided information is inconsistent and leads to many questions regarding the biology of SORT1 in the metabolism of lipids. This enigma might be explained through revealing the complicated, bidirectional nature of sortilin; however, the exact mechanisms underlying observed phenomena are still mainly unknown [72].

Second, regarding the lipid metabolism, SORT-1 is suspected to be involved in the progression of Alzheimer’s disease and alteration of the glucose metabolism and, in this manner, can be considered as a risk factor for manifestations of DM symptoms [73]. In 2017, Oh et al. [74] demonstrated that SORT1 levels could be considered as a putative biomarker for coronary disease and DM. In murine models, it has been shown that overexpression of sortilin was independently associated with increased levels of PCSK9 and decreased LDL-R activity [75]. On the other hand, Hu et al. [76] verified this crosstalk by showing that PCSK9 concentrations were independently linked to sortilin levels, and interestingly, their correlation was influenced by the use of statin therapy.

### Sortilin and Psoriasis

The crucial and multidirectional role of apoptosis in psoriasis is well known and has been confirmed in numerous studies [77]. To the best of our knowledge, there is no data available from experiments directly investigating the role of sortilin in psoriasis. The biological foothold that makes SORT1 relevant for psoriatic studies is the presence of p75 neutrophin receptor (p75NTR), which is a member of the TNF-receptor family. SORT1 mutation that resulted in decreased levels of the protein led to p75NTR upregulation and inconsequent impacted regulatory mechanisms of apoptosis in neurons and keratinocytes [78]. Additionally, it has been shown that sortilin acts as a co-receptor for P75NTR [79]. Truzzi et al. [78] was the first who assessed the expression of sortilin in the psoriatic epidermis and revealed its significantly elevated levels. Moreover, the scientists evaluated the ability of activated p75NTR to mediating apoptosis in pro-nerve growth factor (pro-NGF)-dependent manner in keratinocytes. Interestingly, they proved that p75NTR, along with sortilin, augmented the pro-apoptotic role of pro-NGF in human keratinocytes and showed that p75NTR is decreased in psoriatic lesions. Taken together, the given data indicate that sortilin might play some protective role in apoptotic processes in psoriasis; however, further investigations are needed [78]. The concept of the interplay between psoriasis, ANGPL8, and sortilin is shown in Figure 4.

## 5. Cholesteryl Ester Transfer Protein (CEPT)

Cholesterol ester transfer protein (CEPT, plasma lipid transfer protein) belongs to the family of the lipid transfer/lipopolysaccharide-binding proteins. Chemical analysis showed that CEPT is a hydrophobic glycoprotein, mainly secreted by the liver, which was initially discovered in 1977 by Barter et al. [80]. It has been proven that CEPT is expressed profusely in macrophages [81]. The fundamental function of CEPT is to promote the transfer of cholesteryl esters (CE) chains from the HDL fraction to the proatherogenic non-HDL fraction, such as VLDL and LDL. The outcome of CEPT role is the increase of LDL fraction and the contrary effect on HDL concentration. Increased HDL levels in plasma are associated with a lower risk of coronary artery disease (CAD) [82]. So far, two approaches have been presented to explain the CEPT mechanisms of action. The first one is called a shuttle model, in which CEPT attaches transiently to lipoprotein and promotes the exchange of cholesteryl esters and triglyceride between these two molecules [83]. The second model, called a tunnel model, in which CEPT interacts with HDL, LDL, or VLDL, forming a triple-complex and subsequently creates the hydrophobic tunnel between lipoprotein molecules. As the results of this assembly procedure, cholesteryl ester is transferred from HDL to lower density proteins, and the activity of CEPT also is regulated by the HDL and LDL levels [83]. The levels of CEPT seem to depend on a plethora of different factors, such as consume of food rich in cholesterol, waist circumference, hormone metabolism, and prescribed medications [81].

It has been proven that the level of CEPT is increased significantly in patients with dyslipidemia [81]. In the basic science, when using animal models, the scientists observed a reduction of HDL levels and increased LDL and VLDL levels after the administration of human-type CEPT to mice. The stimulating report published recently showed that in rabbits treated with CEPT inhibitors, the prevalence of atherosclerosis was significantly lower than in non-treated animals [84]. Unfortunately, in clinical studies, the data are more conflicting, especially regarding the incidence of coronary artery disease, which was studied most widely in the view of CEPT biology. Amer et al. [85] reported a positive association between high serum CETP levels and acute coronary syndrome (ACS) incidence. On the contrary, Martinelli et al. [86] proved that decreased CETP concentrations may be an indicator of heart failure severity and deliver poor prognosis [86]. Girona et al. [87] in 2016 reported that CEPT levels were significantly elevated in patients with CMS, thus for the first time suggesting the existence of direct involvement of CEPT into the course of CMS. Moreover, the association between PCSK9 and CEPT levels were investigated and increased PCSK9 level corresponded with higher CEPT activity [87].

It has been widely proven that CEPT is correlated with higher blood levels of HDL, lower LDL cholesterol, and increased risk of coronary heart disease suggested that pharmacological inhibition of CEPT may be favorable in the treatment and prevention of CVD. Currently, four concepts of CEPT inhibitors have achieved the third phase of their clinical trials—torcetrapib, dalcetrapib, evacetrapib, and anacetrapib. The most recently discovered CEPT inhibitor is TA-8995 (also known as AMG-8995) that shows improved pharmacodynamic and pharmacokinetic properties [84]. Therapy with torcetrapib significantly elevated the level of HDL by 70%, decreased levels of LDL by 25%, and seems to normalize lipid metabolism [88]. On the other hand, in patients treated with torcetrapib, a significant increase in cardiovascular events has been reported by the sponsor of the trial [88]. Dolcetrapib increased HDL levels by 30%, without any significant and favorable effect on the level of LDL and had no impact on the occurrence of cardiovascular events [84]. Another clinical trial—ACCELERATE—showed that evacetrapib reduced LDL levels by approximately 30% and doubled the concentration of HDL, however, no benefits described as a reduction of cardiovascular events were found [89]. The safety and efficacy of TA-8995 were evaluated during a 12-week trial. Treatment with TA8995 reduced LDL levels by 45% and increased HDL levels by 180% without any life-threatening complications [90]. These contradictory results of the clinical trials give a strong scientific background for the potential role of CEPT inhibition for prevention of lipid metabolism disruptions and further CMS.

### Cholesteryl Ester Transfer Protein and Psoriasis

To date, there are no published papers directly investigating the role of CEPT in psoriasis. However, the data from the literature provided in this section highlight the importance of CEPT in CMD. The interplay between CMS and psoriatic pathophysiology emphasizes the need to explore the role of this protein in the pathogenesis of psoriasis and its comorbidities.

## 6. Conclusions

Numerous studies have outlined that psoriasis and cardiometabolic diseases share many immunological pathways, and their complicated interplay leads to the development of psoriatic march. Today, chronic inflammation, disruption in the lipid balance, and disturbances in glucose metabolism seem to play a crucial role in tightening the reciprocated relationship between psoriasis and CMS. The abnormalities in proteins levels and activities described in this paper lead to the occurrence of divergent metabolic shifts that, taken together, highly impact the progression of pro-psoriatic and pro-cardiometabolic phenotypes. Due to the increased mortality in patients with psoriasis from cardiovascular disorders and its complications, searching for newer indicators seems to be a prerequisite part of modern dermatology. Based on that, in the future dermatologist would be able to use therapeutic strategies to manage the skin problems and also reduce the risk of cardiometabolic comorbidities. So far, in our previous study, we proved that PCSK9 has to be considered a novel biomarker of psoriasis, and further, we have proven that methotrexate should be the treatment of choice in patients with an elevated PCSK9 concentration before treatment [48].

## Figures and Tables

**Figure 1 ijms-21-03682-f001:**
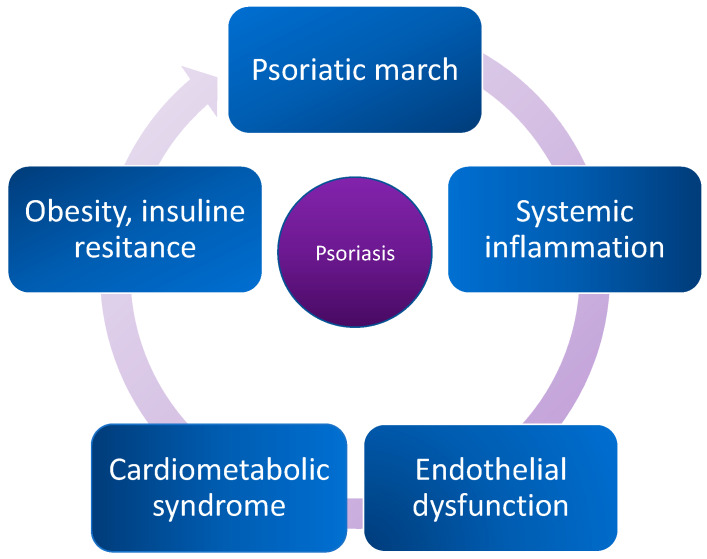
The “psoriatic march”: an old-new concept of how psoriasis may drive cardiovascular comorbidity.

**Figure 2 ijms-21-03682-f002:**
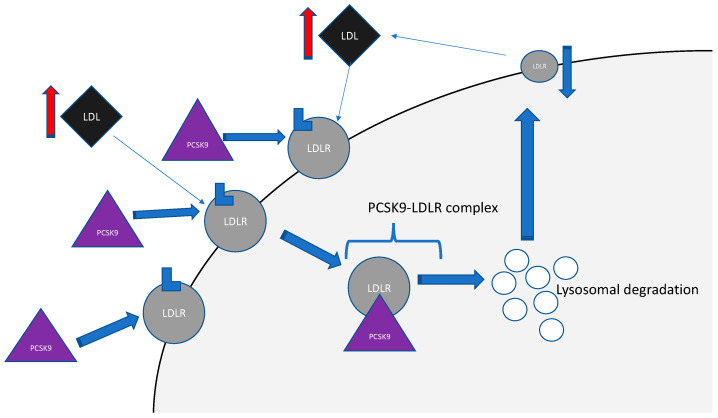
Secreted PCSK9 binds to LDLR on the liver surface and mediates the lysosomal degradation of the complex formed by PCK9-LDLR, leading to decreased density of LDLR on the hepatocyte surface and further to increased LDL level in serum. Abbreviations: LDL—low-density lipoprotein; PCSK9—proprotein convertase subtilisin/kexin type 9; LDLR—low-density lipoprotein receptor.

**Figure 3 ijms-21-03682-f003:**
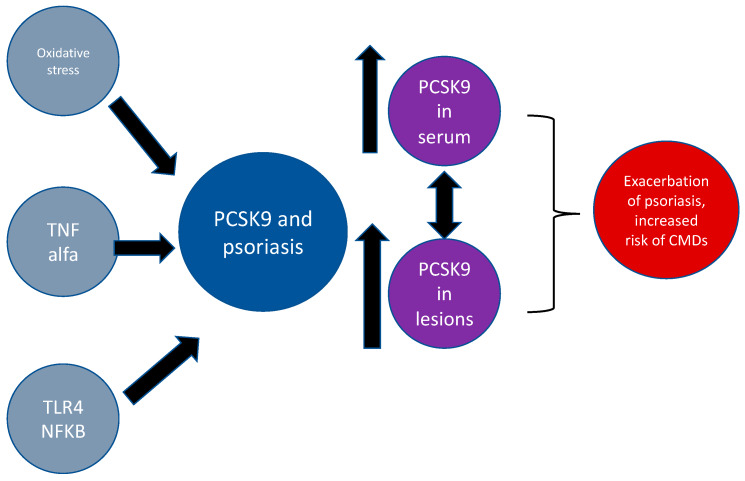
The putative connections between PCSK9 and psoriasis. Abbreviations: PCSK9-proprotein convertase subtilisin/kexin type 9; TNF alfa—tumor necrosis factor alfa; TLR4—toll-like receptor 4; NFKB—nuclear factor kappa-light-chain-enhancer of activated B cells.

**Figure 4 ijms-21-03682-f004:**
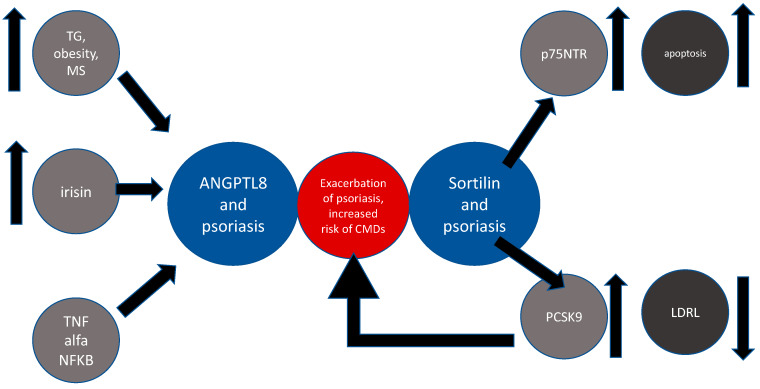
Common links between ANPTL8, sortilin, and psoriasis. Abbreviations: TG—triglycerides; TNF-α—tumor necrosis factor alfa; NF-kB—nuclear factor kappa-light-chain-enhancer of activated B cells; ANGPTL8—angiopoietin-like protein 8; p75NTR—p75 neutrophin receptor; LDLR—low-density lipoprotein receptor; MS—metabolic syndrome.

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
