# Peer review of "Proprotein Convertase Subtilisin/Kexin Type 9, Angiopoietin-Like Protein 8, Sortilin, and Cholesteryl Ester Transfer Protein—Friends of Foes for Psoriatic Patients at the Risk of Developing Cardiometabolic Syndrome?"

_ijms, 2020, doi:10.3390/ijms21103682_

Round 1
Reviewer 1 Report
none
Author Response
Thank you very much for your valuable time and reviewing the manuscript.
Best regards,
Julita Krahel
Reviewer 2 Report
This review article addresses an important subject. The link between psoriasis and cardiometablic syndrome is discussed. The authors focus on four proteins that play a role in this articulation. This choice is justified and should be of interest to skin as well as metabolism researchers
Comments:
1) The structure of the paper has to be better determined. There is a chapter 1, the Introduction, then there is a series of sub-chapters that are not numbered. Finally, there is a short last chaper 5, the Conclusions. There is no numbering in between, one jumps from 1 to 5. This has to be fixed.
2) The sentence at the top of page 5 starting with "Interesting observations ..." is not clear. It should be re-written.
3) There should be a reference given at the end of the sentence, on page 7, starting with "Recently published our paper, confirmed that PCSK9 seems ..." Also note that there is an English language problem with this sentence.
4) Page 8, lower half. A reference is needed at the end of the sentence "To our knowledge, the paper published in 2020 by our team was the first concerning directly ANGPTL8 level and psoriasis severity so far."
5) The English language is of insufficient quality, which reduces the quality of the whole manuscript.
Author Response
Thank you very much for your valuable time and reviewing the manuscript. The manuscript has been reworked according to the suggestions and all changes have been matched with yellow colour.
- The structure of the manuscript has been determined (numbers from 1 to 6)
- The sentence has been changed.
- We have added the reference and re-written the whole sentence.
- The sentence has been removed from the manuscript.
- We have improved the english language in the entire manuscript.
Kindly regards,
Julita Krahel
This manuscript is a resubmission of an earlier submission. The following is a list of the peer review reports and author responses from that submission.
Round 1
Reviewer 1 Report
1) metabolic syndrome sometimes is abbreviated with MetS some Others with MS or MeS
2) in the introduction the molecules are presented with their entire name and abbreviation except: CEPT (add:Cholesteryl ester tranfer protein)
3) English must be revised and make shorter senteces where applicable
4) conclusion is too short. the authors must stressed the fact that a modern dermatologist must be able to deal with cardiometabolic disease. This article is interesting: "Krahel JA, Baran A, Kamiński TW, Maciaszek M, Flisiak I. Methotrexate Decreases the Level of PCSK9-A Novel Indicator of the Risk of Proatherogenic Lipid Profile in Psoriasis. The Preliminary Data.J Clin Med. 2020 Mar 26;9(4). maybe can help to lengthen the conclusion.
Reviewer 2 Report
As indicated in the title, this review article aims to make a link between proprotein convertase subtilisin/kexin type 9, angiopoietin like protein 8, sortilin and cholesteryl ester transfer protein and cardiometabolic conditions and psoriasis.
Potentially, this is a good idea, but the manuscript presents several important weaknesses.
1) It is sometimes difficult to understand what the authors wish to say because the English language is not good enough. Unfortunately, this certainly weakens the manuscript very much.
2) Even if the article is short, it would be important to have a couple of illustrations which enrich the text and make it more understandable. The authors did not make the effort to come up with such a support of their work.
3) Based on the available knowledge there is not so much to say about
- ANGPL8 and psoriasis. The authors write: "To our knowledge, there is no published data concerning directly ANGPTL8 level and psoriasis so far."
- Sortilin and psoriasis. The authors write: "So far, there is lack of data investigating directly the role of sortilin in psoriasis ..."
- Cholesteryl ester transfer protein and psoriasis. The authors write: "To date, there is no published paper investigating directly CEPT in psoriasis."
Therefore, the usefulness of an important part of the review can be questioned.
4) Often, the text lists a series of observations without integrating them and proposing a take home message, which reflects some superficiality and a lack of in depth analysis.
At this stage, there is doubt whether this work reaches the standards of those usually published in IJMS.